# Biomechanical Variability and Usability of a Novel Customizable Fracture Fixation Technique

**DOI:** 10.3390/bioengineering10101146

**Published:** 2023-09-29

**Authors:** Thomas Colding-Rasmussen, Peter Schwarzenberg, Peter Frederik Horstmann, Casper Bent Smedegaard Ottesen, Jorge San Jacinto Garcia, Daniel John Hutchinson, Michael Malkoch, Michael Mørk Petersen, Peter Varga, Christian Nai En Tierp-Wong

**Affiliations:** 1Department of Orthopedic Surgery, Hvidovre University Hospital, Kettegaard Allé 30, 2650 Hvidovre, Denmark; christian.nai.en.tierp-wong.01@regionh.dk; 2AO Research Institute Davos, Clavadelerstrasse 8, 7270 Davos, Switzerland; peter.schwarzenberg@aofoundation.org (P.S.); peter.varga@aofoundation.org (P.V.); 3Department of Orthopedic Surgery, Gentofte Hospital, Gentofte Hospitalsvej 1, 2900 Hellerup, Denmark; peter.horstmann@regionh.dk; 4Department of Orthopedic Surgery, Rigshospitalet, Copenhagen University Hospital, Blegdamsvej 9, 2100 Copenhagen, Denmark; casper.bent.smedegaard.ottesen.01@regionh.dk (C.B.S.O.); michael.moerk.petersen@regionh.dk (M.M.P.); 5Department of Fibre and Polymer Technology, KTH Royal Institute of Technology, Brinellvägen 8, 10044 Stockholm, Sweden; jorgesjg@kth.se (J.S.J.G.); danhut@kth.se (D.J.H.); malkoch@kth.se (M.M.); 6Faculty of Health and Medical Sciences, University of Copenhagen, Blegdamsvej 3B, 2200 Copenhagen, Denmark

**Keywords:** in situ customizable osteosynthesis, patient-specific osteosynthesis, usability, variability, surgical skills

## Abstract

A novel in situ customizable osteosynthesis technique, Bonevolent™ AdhFix, demonstrates promising biomechanical properties under the expertise of a single trained operator. This study assesses inter- and intra-surgeon biomechanical variability and usability of the AdhFix osteosynthesis platform. Six surgeons conducted ten osteosyntheses on a synthetic bone fracture model after reviewing an instruction manual and completing one supervised osteosynthesis. Samples underwent 4-point bending tests at a quasi-static loading rate, and the maximum bending moment (BM), bending stiffness (BS), and AdhFix cross-sectional area (CSA: mm²) were evaluated. All constructs exhibited a consistent appearance and were suitable for biomechanical testing. The mean BM was 2.64 ± 0.57 Nm, and the mean BS was 4.35 ± 0.44 Nm/mm. Statistically significant differences were observed among the six surgeons in BM (*p* < 0.001) and BS (*p* = 0.004). Throughout ten trials, only one surgeon demonstrated a significant improvement in BM (*p* < 0.025), and another showed a significant improvement in BS (*p* < 0.01). A larger CSA corresponded to a statistically significantly higher value for BM (*p* < 0.001) but not for BS (*p* = 0.594). In conclusion, this study found consistent biomechanical stability both across and within the surgeons included, suggesting that the AdhFix osteosynthesis platform can be learned and applied with minimal training and, therefore, might be a clinically viable fracture fixation technique. The variability in BM and BS observed is not expected to have a clinical impact, but future clinical studies are warranted.

## 1. Introduction

Bone fractures are considered an increasing global health burden and a substantial cause of years lived with disability [1]. If surgical fixation of a fracture is required for stability and bone healing, conventional metal-based implants such as plates, screws, nails, rods, and wires are most commonly used [2]. The traumatic fracture patterns occurring in various parts of the skeleton are often similar in morphology, which is why most metal-based implants are predesigned to fit a specific fracture type [3]. However, in some cases, fractures can be more complex and multi-fragmented due to severe trauma or a fragile osteoporotic bone structure and might, therefore, benefit from a more customizable fracture fixation solution [4,5,6]. Moreover, metal implants, particularly when inserted near tendons in phalanx fractures, tend to promote adhesions in the surrounding soft tissue which can cause functional impairment [7,8,9].

A novel customizable fracture fixation solution (Bonevolent™ AdhFix, Biomedical Bonding AB, Stockholm, Sweden) has recently been proposed as an alternative to conventional metal-based implants in fracture management [10]. With this technique the surgeon builds a plate-like structure from a light-cured composite that is anchored to the bone with conventional metal screws, creating a patient-specific fracture fixation solution designed in situ. 

The AdhFix osteosynthesis platform has previously been biomechanically studied in various conformations in cyclic loading, bending and torsion in ex vivo porcine phalanx—and ovine metacarpal fracture models in accordance with established biomechanical test recommendations and has shown promising biomechanical results when compared to conventional metal plates and Kirschner wires [10,11,12,13]. Overall, the AdhFix platform has demonstrated superior stiffness and inferior max bending moment when compared to a similarly sized conventional locking plate in reduced ex vivo ovine transverse fractures in 4-point bending. However, studies suggest that the AdhFix platform provides sufficient biomechanical support to treat phalanx fractures, even though it does not provide the same ultimate bending moment as a corresponding metal plate [10,11]. Furthermore, in torsional biomechanical tests, AdhFix demonstrated superior stiffness and comparable maximum torque when compared to small fragment locking plates [11].

The advantage of this innovative technique is that it enables surgeons to create a fracture-specific osteosynthesis in situ, potentially optimizing bone healing [14,15,16]. However, the flexibility inherent in this approach also means that variations in size, shape, and biomechanical properties between each osteosynthesis are to be expected. Up until now the biomechanical studies of the AdhFix technique have been analyzed using only a single biomechanical researcher or a single surgeon to eliminate potential variations and allow for a comparison between different osteosynthesis techniques and AdhFix [10,11]. 

However, studies show that both inter- and intra-surgeon variations in biomechanical performance occur when placing conventional metal screws as well as variations between surgeons and non-medically trained biomechanical researchers [17,18]. This highlights the importance of investigating the biomechanical variability between and within surgeons to fully biomechanically characterize new osteosynthesis platforms. 

Furthermore, orthopedic surgeons are taught how to use basic metal-based implants and corresponding tools in fracture management early in their career in clinical training, simulation training and at courses facilitated by the Arbeitsgemeinschaft für Osteosynthesefragen (AO) or other institutions. Consequently, the surgeon’s lack of experience with a novel fracture fixation platform such as AdhFix might lead to application issues and a corresponding lack of biomechanical stability [19,20,21]. Previous studies have not explored whether and how the biomechanical stability of the AdhFix osteosyntheses platform varies between and within users. Given that the AdhFix-technique demands an alternative and customizable surgical approach compared to that of conventional metal plating, it is important to investigate if the osteosynthesis platform is biomechanically viable when applied by new users. Accordingly, the purpose of this study was to investigate on an experimental level the usability and variability in the biomechanical stability of the AdhFix osteosynthesis technique.

## 2. Materials and Methods

AdhFix is composed of trifunctional allyl and thiol triazine-trione monomers (1,3,5-triallyl-1,3,5-triazine-2,4,6 (1H,3H,5H)-trione and 1,3,5-tris (3-mercaptopropyl)-1,3,5-triazinane-2,4,6-trione, respectively), a photo-initiator (diphenyl (2,4,6-trimethylbenzoyl) phosphine oxide (TPO)) and hydroxyapatite. It is cured with a handheld high-energy visible (HEV) light lamp and anchored to the bone with conventional metal screws [10]. In this study, the AdhFix osteosynthesis platform was applied on a well-reduced complete transverse osteotomy with two titanium 1.5 mm unicortical bone screws (Medifit Devices LLP, Ahmedabad, India) placed in each fragment with a span of 5 mm between each screw and 5 mm from the fracture line. The AdhFix composite was placed around each screw, over the fracture gap and cured, after which two subsequent layers of AdhFix were added and cured stepwise covering the entire desired rectangular patch size of 25 mm × 6 mm as marked on the bone. All screws were pre-inserted in the drill holes and left protruding approximately 3 mm (Figure 1).

Synbone^®^ generic bone model (PR0347: Ø 12 mm, wall 3 mm, SYNBONE AG, Zizers, Switzerland) was used in this study. Each fragment had a length of 30 mm with two 1.0 mm drill holes positioned longitudinally, with a 5 mm span and a 5 mm distance from the fracture line (Figure 2). To ensure stability during osteosynthesis creation and biomechanical testing, 1 mm was milled on the bottom surface of the synthetic bone fragments, creating a flat section. These fragments were meticulously prepared at a professional machine shop to ensure optimal accuracy. Three-dimensional (3-D) printed plastic bone fragment holders were created to achieve equal reduction and stability on each sample while performing the procedure. 

Six surgeons with various experience levels were recruited in this study: a consultant hand surgeon (surgeon 1), a specialist hand surgeon (surgeon 2), a specialist arthroplasty surgeon (surgeon 3), a 1st-year resident (surgeon 4), an intern surgical novice (surgeon 5) and a 1st-year resident with prior AdhFix osteosynthesis experience (surgeon 6). Each surgeon reviewed an identical instruction manual for performing the AdhFix-procedure (Appendix A) after which each surgeon performed a test-osteosynthesis supervised by an experienced AdhFix user before the test results were recorded. Each surgeon then performed ten consecutive osteosyntheses without supervision. Each sample was tested immediately after the completion of the osteosynthesis. 

A custom-made four-point bending fixture was employed with a 15 mm span on the bottom surface and a 44 mm span on the top surface. The fixture was mounted on a universal testing machine (Inspekt Duo 5, Hegewald & Peschke, Nossen, Germany) with a 5 kN load cell and a quasi-static compression rate of 3 mm/min was applied (Figure 3). Maximum load at failure was evaluated as the largest force value before failure and the corresponding maximum bending moment (BM) was calculated from the bending moment arm determined from the fixture dimensions. Bending stiffness (BS) was calculated from the moment versus displacement curve between 25–75% of the maximum bending moment [11]. 

The main outcome variables measured were BM (Nm) and BS (Nm/mm). The thickness of the AdhFix osteosynthesis patches was measured at the thickest point on each side of the fracture after the mechanical test was complete (mean was calculated), while the width was measured at screw level on each side of the fracture (mean was calculated). Subsequently, the cross-sectional area (CSA) was calculated (width × thickness; in mm^2^). The time required to complete the osteosynthesis was measured for each sample and surgeon. 

Descriptive statistics and a linear mixed effect model were used to calculate and describe differences in BM and BS between and within surgeons. The coefficient of variance (CV) in percent was calculated as the ratio of the standard deviation (SD) to the mean. The variable ‘surgeon’ was applied as a random factor with and without the inclusion of the variable ‘cross-sectional area’ or ‘trial’ as a fixed covariate. The time required to complete the osteosynthesis was not included in the statistical model since the fracture was pre-reduced and screws were inserted in advance. Bonferroni correction was applied for post-hoc tests for pairwise comparisons among the surgeons. Linearity and normality were assessed in exploratory statistics using scatter plots and QQ-plots. Values are reported as mean ± standard deviation unless otherwise stated. *p* values < 0.05 were considered statistically significant and the corresponding 95% confidence intervals (CI) were applied when relevant. IBM (Corp. Armonk, NY, USA) SPSS version 27 was used for statistical analysis. This is an explorative study and power calculations were not conducted before execution.

## 3. Results

Across all surgeons and trials, BM was 2.64 ± 0.57 Nm, and BS was 4.35 ± 0.44 Nm/mm. The overall coefficient of variance was 21.6% in BM and 10.1% in BS, respectively. The average time to complete the osteosynthesis across all surgeons was 12:41 ± 02:37 min: sec, with an overall coefficient of variation of 20.7% (Table 1). All AdhFix patches created were observed to be symmetrical and exhibited a similar appearance (Figure 4). No AdhFix constructs were excluded due to immediate plate failure or poor construction.

### 3.1. Inter-Surgeon Variation 

Surgeon 3 had the highest BM (3.23 ± 0.45 Nm), followed by surgeon 5 (3.03 ± 0.30 Nm), surgeon 1 (2.90 ± 0.55 Nm), surgeon 6 (2.49 ± 0.29 Nm), surgeon 4 (2.23 ± 0.30 Nm) and then surgeon 2 (2.01 ± 0.34 Nm; Table 1 and Figure 5). An overall statistically significant difference was found in BM between the six surgeons recruited for this study (*p* < 0.001). The post-hoc tests revealed that surgeon 3 and surgeon 5 created AdhFix fixations with higher BM than surgeons 2, 4 and 6 while surgeon 1 had a higher BM than surgeon 2 and surgeon 4, respectively (*p* < 0.038 for all comparisons stated). The statistical model showed that a higher average CSA was associated with a higher BM: 0.145 Nm per 1 mm^2^ (CI: 0.08–0.21), *p* < 0.001. When adjusted for CSA, a statistically significant difference in BM between the surgeons remained (*p* = 0.02). However, only surgeon 3 had a statistically significantly higher BM than surgeon 2 in the post hoc test with a mean estimated difference of 0.7 Nm, (CI: 0.10–1.29), *p* < 0.001. All other comparisons were not statistically significant. 

In BS, surgeon 6 had the highest mean (4.68 ± 0.40 Nm/mm), followed by surgeon 5 (4.54 ± 0.59 Nm/mm), surgeon 1 (4.47 ± 0.36 Nm/mm), surgeon 3 (4.20 ± 0.21 Nm/mm), surgeon 2 (4.16 ± 0.33 Nm/mm) and surgeon 4 (4.06 ± 0.32 Nm/mm; Table 1 and Figure 6). A statistically significant difference in BS was observed between the six surgeons overall, *p* = 0.004. The post-hoc tests showed only a statically significant difference between surgeon 6 and surgeon 4 with an estimated mean difference of 0.62 Nm/mm, (CI: 0.08–1.15), *p* = 0.012. No statistically significant relationship was found between CSA and BS: −0.021 Nm/mm pr. mm^2^, (CI = −0.01–0.06), *p* = 0.594. When adjusting for the CSA, surgeon 6 remained statistically significantly higher in BS than surgeon 4 with a mean estimated difference of 0.63 Nm/mm (CI: 0.08–1.17), *p* = 0.012.

### 3.2. Intra-Surgeon Variation

All surgeons completed 10 trials except for surgeon 3 who only completed 9 trials due to a technical issue. Linear estimation across all trials showed a modest tendency towards higher BM with more completed trials, except for surgeon 2 where a modest negative trend was observed (Figure 7). However, only surgeon 1 exhibited a statistically significant interaction between the number of trials and increasing BM with and without adjusting for the CSA (*p* < 0.025 in both models). For BS, a similar trend towards higher values with repeated trials was observed, except for surgeon 4, where a slight negative trend was noted (Figure 8). Only surgeon 5 demonstrated a statistically significant interaction between trials and increasing BS with and without adjusting for the CSA of the AdhFix patch (*p* < 0.01 in both models). 

## 4. Discussion

In this study, we conducted a biomechanical evaluation to assess intra- and inter-surgeon variability and usability of a novel customizable osteosynthesis technique, AdhFix. This experimental study included a limited dataset and provided few observations overall. 

All AdhFix constructs created were similar in appearance and suitable for biomechanical tests. Consequently, we found that all surgeons, regardless of their previous experience level, were able to consecutively create stable fracture fixation constructs in 12:41 ± 2:37 min after reading the instruction manual (Appendix A) and conducting one supervised osteosynthesis, indicating that the AdhFix technique is simple and feasible. However, basic surgical skills such as fracture reduction, screw hole positioning and drilling were not assessed in this study, though these variables might impact the biomechanical stability, healing and in turn inter-surgeon variation in actual in vivo fracture management [18,19,22]. 

We observed an overall modest trend in improving biomechanical stability across consecutive trials. The finding of a statistically significant interaction between the number of trials and BM in surgeon 1 (consultant hand surgeon) and BS in surgeon 5 (surgical novice), respectively, might be due to the small sample size and variation in this experiment. Consequently, we do not know if a plateau exists in the biomechanical stability of the AdhFix fracture solution. However, given the intra-surgeon variation and the very modest trend observed we do not suspect that surgeons will improve the biomechanical stability across several repetitions. This further indicates that becoming proficient at the AdhFix technique does not require extensive training.

In previous studies, the AdhFix osteosynthesis technique has been suggested to be a feasible solution in phalanx fracture management [10,11]. In this study, we found a mean BM of 2.64 ± 0.57 Nm, which corresponds to a load of 364.1 ± 78.6 N applied to the 4-point bending fixture that was utilized in this experiment. Though these numbers are not directly comparable due to different loading modalities, they provide an estimate of the internal loads within the bone model. This value exceeds what has previously been reported in a biomechanical evaluation of loads occurring in phalanges during a variety of daily living activities; up to 34.8 ± 1.6 N [23]. However, studies examining actual internal loads in corresponding bone fracture models are warranted to further quantify if the biomechanical stability of AdhFix is sufficient in the treatment of hand fractures.

In this study, we found a statistically significant interaction between the CSA and BM. This finding collaborates findings of another study investigating the relationship between AdhFix plate thickness and biomechanical performance indicating that increasing plate width also contributes to increased biomechanical performance [10,14]. When adjusted for CSA in this study, only the surgeons with the highest and lowest BM, surgeon 2 and surgeon 3, respectively, remained statistically significantly different from each other. This is important knowledge for future clinical applications of AdhFix in that the surgeons must be aware of this association between plate CSA and bending moment performance to fully utilize the customizability of this technique.

In BS, we observed a statistically significant difference only between the surgeon with the highest mean (surgeon 6) and the surgeon with the lowest mean (surgeon 4) with and without adjusting for the CSA. Interestingly, the CSA did not significantly impact the bending stiffness. Consequently, bending stiffness appeared to be less sensitive to variations between surgeons and the corresponding CSA than BM. This could potentially be an advantageous attribute to the AdhFix solution since a known and relatively constant fixation stiffness, that is less responsive to patch morphology variations, makes it easier to achieve a sufficiently stiff construct while maintaining a relatively thin patch. This, in turn, could improve aesthetics and reduce skin- and soft tissue irritation from underlying osteosynthesis materials [16].

Comparing the variation in BM and BS of AdhFix to that of metal plates is difficult due to a lack of studies investigating inter- and intra-surgeon variation concerning biomechanical properties. However, a recently published study presented a BM of 19.51 ± 2.24 Nm and BS of 8.69 ± 1.16 Nm/mm in an ex vivo ovine metacarpal transverse reduced fracture model osteosynthesized with a 5-hole 1.5 DePuy Synthes metal plate and 4 bicortical locking screws [11]. This corresponds to a variation coefficient of 11.48% in BM and 13.34% in BS which is comparable to the variability found in this study (Table 1). This indicates that, although AdhFix is a customizable fracture solution, its variability is comparable to that of a standard pre-made metal plate when constructed in a standard defined size and shape.

A potential explanation of the intra- and inter-surgeon variability that was observed in this study could be a variation in screw insertion technique. In the AdhFix platform, each screw is inserted carefully ‘two-finger tight’ to allow the composite to remain under the screw head and function as anchorage to the bone [10]. Screw torque has previously been shown to vary between and within surgeons and may, therefore, contribute to the variability found in this study [24]. Furthermore, the concept of ‘two-finger tight’ has been shown to be associated with significant variation in torque, bone stripping and lack of reproducibility when screws are placed [25]. To reduce the variation of screw torque, the use of an augmented screwdriver indicating when torque is optimal has been shown to improve the optimal tightening of screws while also reducing bone stripping [26]. This technique might be an adjuvant in the AdhFix platform to potentially reduce the variability.

We observed no clear association between surgical experience and biomechanical performance of the AdhFix constructs. In BM we found only a statistically significant difference between surgeon 3 (specialist arthroplasty surgeon) and surgeon 2 (specialist hand surgeon) after adjusting for CSA, indicating that prior orthopedic training did not affect BM. In BS, surgeon 6, an experienced AdhFix user had the highest value, though it was only statistically significantly different from surgeon 4 who had the lowest mean bending stiffness. This might be an indication that experience in the AdhFix technique could improve biomechanical stability. This could be due to a better understanding and tactile feeling of when the screw is tightened sufficiently to allow enough composite to anchor the material to the bone while ensuring sufficient screw-bone stability, which is an important aspect of plate stability [27].

Though data showed a statistically significant difference between the surgeons in BS and BM it is uncertain if these relatively subtle variations in biomechanical stability would impact in vivo fracture stability and fracture gap motion, and therefore, whether they would influence bone healing. With an overall standard deviation of 0.57 Nm and 0.44 Nm/mm in BM and BS, respectively, we do not hypothesize that these variations between surgeons would have a clinical impact on bone healing. However, it would depend on the current internal loads that a specific fracture is exposed to during rehabilitation exercises and in turn, the stress and strain occurring in the osteosynthesis construct [11,13,22]. Future studies that analyze topology optimization of the AdhFix plate morphology and screw position concerning stress, strain and bone healing in specific fracture types are warranted to be able to further characterize the potential of this osteosynthesis technique [28].

This study is not without limitations. First, we only examined fracture stability in a four-point bending loading setup. During actual in vivo motions, a bone is typically exposed to various loading modalities including bending, compression, torsion, and cyclic loading [29]. Second, we used a synthetic bone model to standardize the biomechanical test setup though human bone would provide a more realistic platform. Completely identical bone fragments could only be achieved with a synthetic bone model. Third, in this study, the circumstances to create a stable fracture fixation were ideal, with no soft tissue, a dry surface, and a perfectly reduced fracture. In real-life surgery, particularly in complex fractures, a more experienced surgeon might create a more stable fracture fixation due to an improved reduction and fragment alignment [30]. This could not be shown with this study setup. 

## 5. Conclusions

In this study, we found that the AdhFix osteosynthesis technique was applicable and reproducible regardless of prior surgical experience. Further studies are warranted to quantify if the actual variation in BM and BS will have a clinical impact on fracture stability and bone healing. Accordingly, AdhFix might be an adjuvant in fracture management as it provides the surgeon with an easily customizable yet reproducible and biomechanical stabile fracture fixation solution.

## Figures and Tables

**Figure 1 bioengineering-10-01146-f001:**
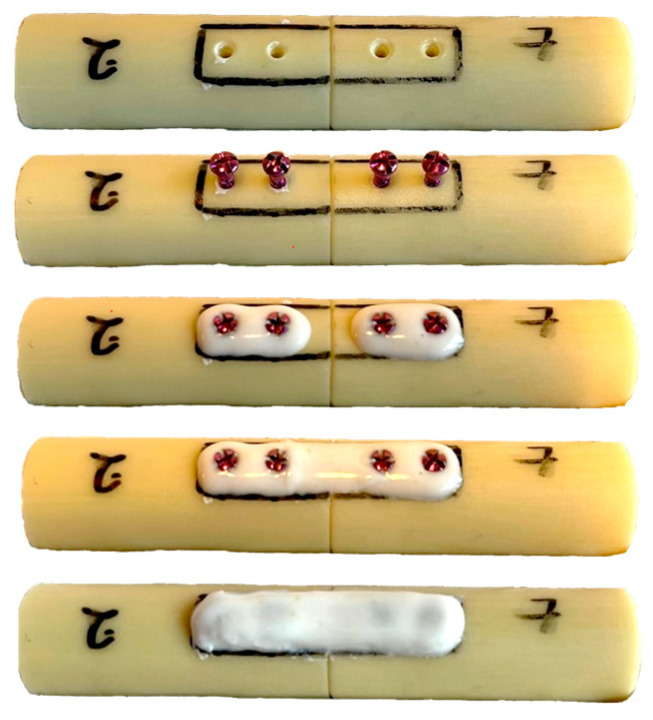
Illustration of the stepwise application of the AdhFix-technique.

**Figure 2 bioengineering-10-01146-f002:**
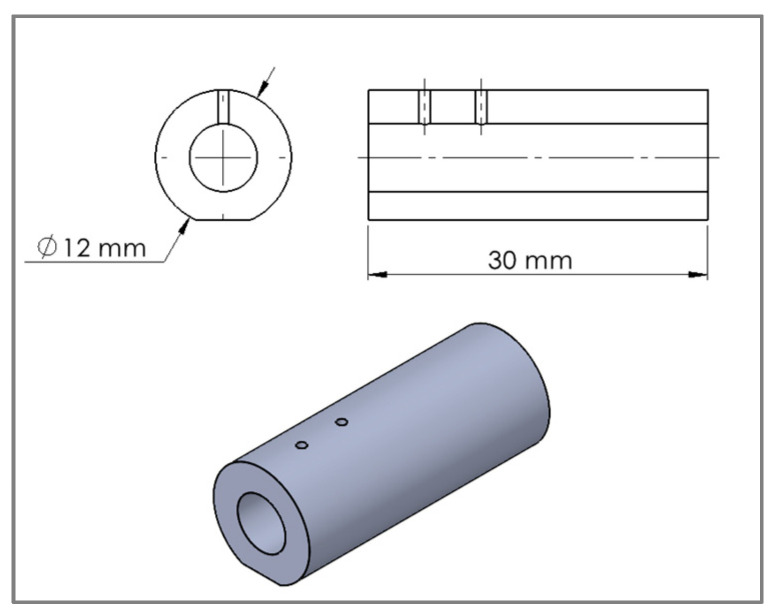
Illustration of the synthetic bone model (one fragment shown).

**Figure 3 bioengineering-10-01146-f003:**
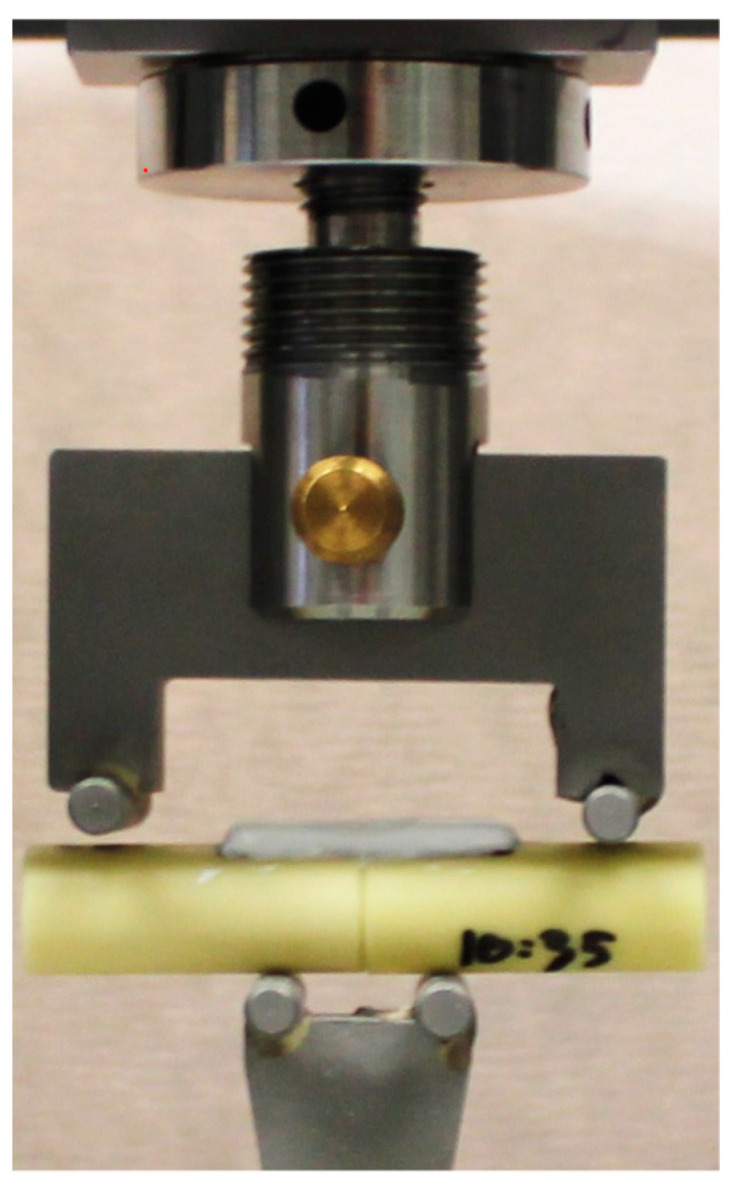
Illustration of the four-point bending fixture and example of AdhFix fixation on the synthetic bone fracture model.

**Figure 4 bioengineering-10-01146-f004:**
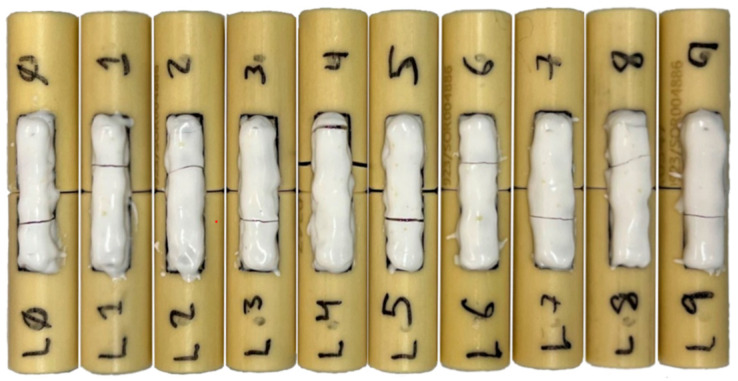
Example of samples after four-point bending to failure.

**Figure 5 bioengineering-10-01146-f005:**
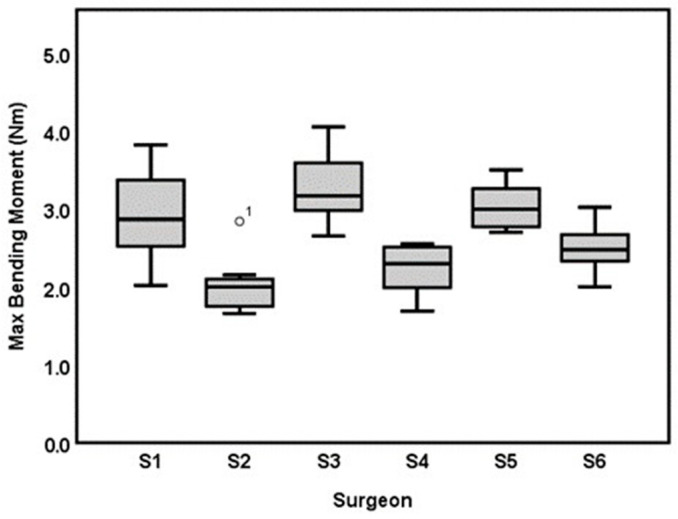
Boxplot of max bending moment (Nm) for each of the six surgeons included in the study (S1–S6). One outlier illustrated in S2.

**Figure 6 bioengineering-10-01146-f006:**
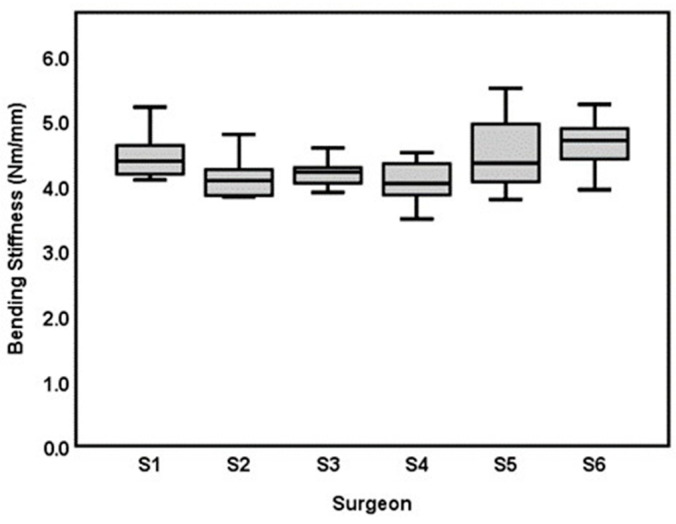
Boxplot of bending stiffness (Nm/mm) for each of the six surgeons included in the study (S1–S6).

**Figure 7 bioengineering-10-01146-f007:**
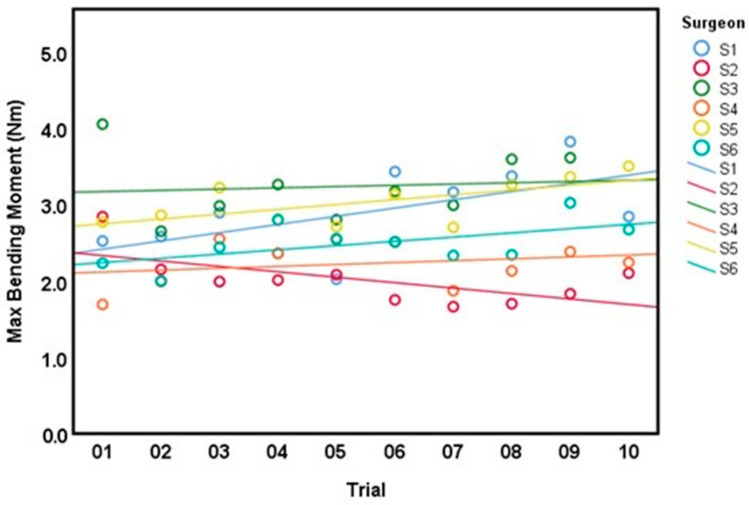
Linear model of max bending moment (Nm) for all surgeons (S1–S6) across 10 consecutive trials (1–10).

**Figure 8 bioengineering-10-01146-f008:**
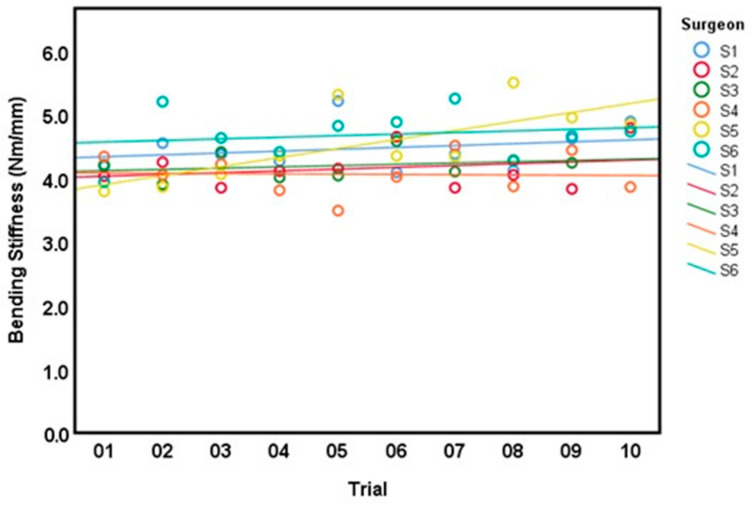
Linear model of bending stiffness (Nm/mm) for all surgeons (S1–S6) across 10 consecutive trials (1–10).

**Table 1 bioengineering-10-01146-t001:** Max load at failure (Nm), bending stiffness (Nm/mm), time to complete osteosynthesis (min:sec) and cross-sectional area (mm^2^) between the six surgeons included in the study and in total across all surgeons and trials.

Surgeon	Bending Moment (Nm)Mean ± SD (CV)	Bending Stiffness (Nm/mm) Mean ± SD (CV)	Cross Sectional Area (mm^2^) Mean ± SD	Time (min:s), Mean ± SD
Surgeon 1: Consultant hand surgeon	2.90 ± 0.55 (19%)	4.47 ± 0.36 (8%)	12.17 ± 2.02	13:51 ± 00:27
Surgeon2: Specialist hand surgeon	2.01 ± 0.34 (18%)	4.16 ± 0.33 (8%)	9.24 ± 1.52	08:24 ± 00.25
Surgeon 3: Specialist arthroplasty surgeon	3.23 ± 0.45 (14%)	4.20 ± 0.21 (5%)	12.88 ± 1.36	11:22 ± 00.34
Surgeon 4: 1st-year Resident	2.23 ± 0.30 (13.5%)	4.06 ± 0.32 (7.8%)	9.14 ± 1.12	13:32 ± 01:05
Surgeon 5: Intern (surgical novice)	3.03 ± 0.30 (10%)	4.54 ± 0.59 (13%)	12.77 ± 1.18	16:29 ± 01:08
Surgeon 6: 1st-year resident (AdhFix experienced)	2.49 ± 0.29 (11.6%)	4.68 ± 0.40 (8.5%)	9.75 ± 0.84	12:24 ± 00:40
Total trials (N = 59)	2.64 ± 0.57 (21.6%)	4.35 ± 0.44 (10.1%)	10.96 ± 2.12	12:41 ± 02:37

## Data Availability

The raw data from the biomechanical tests are available in the following public reposit: http://doi.org/10.5281/zenodo.8304806. Accessed 31 August 2023.

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
