# Peer review of "Biomechanical Variability and Usability of a Novel Customizable Fracture Fixation Technique"

_bioengineering, 2023, doi:10.3390/bioengineering10101146_

Round 1

Reviewer 1 Report

Ref. No.: bioengineering-2616626

Subject: Decision on Manuscript: Variability and Usability of a Novel Customizable Fracture Fixation Technique: A Biomechanical Evaluation

Journal: Bioengineering

Dear Editor,

I would like to thank you for the invitation to collaborate to review process of article “Variability and Usability of a Novel Customizable Fracture Fixation Technique: A Biomechanical Evaluation”. I recommend that is necessary a major revision of manuscript. Some comments are described below:

English should be improved in all manuscript.

The title should be re-written to explore the main idea of this manuscript that is the statistical analysis based on reproducibility of handling of different surgeons in biomechanical analysis of AdhFix.

Abstract: The abstract should be re-written; it is very confusing. In addition, the novelty of this research is not so clear in the abstract.

Introduction: In general, the novelty of this research is not so clear. In addition, the authors should be emphasized the difference of this research from others performed with the same materials, as they afformed that “Up until now the biomechanical studies of the AdhFix technique have been analyzed 62 using only a single biomechanical researcher or a single surgeon to eliminate potential 63 variations and allow for a comparison between different osteosynthesis techniques and 64 AdhFix [7, 8]”.

Materials and methods:

“AdhFix is composed of trifunctional allyl and thiol triazine-trione monomers, a 78 photo-initiator (diphenyl (2,4,6-trimethylbenzoyl) phosphine oxide (TPO)) and hydroxy- 79 apatite. It is cured with a handheld high-energy visible (HEV) light lamp and anchored to 80 the bone with conventional metal screws [7].” The chemical structure of AdhFix should be included in the manuscript, this information is very interesting and explain some important aspects of mechanical behavior.

“The fixture was mounted on a universal  testing machine (Inspekt Duo 5, Hegewald & Peschke, Nossen, Germany) with a 5 kN load cell and a quasi-static compression rate of 3 mm/min (Figure 3).” Specific standard used in mechanical test should be mentioned.

“Six surgeons with various experience levels were recruited in this study: A consultant hand surgeon (surgeon 1), a specialist hand surgeon (surgeon 2), a specialist  arthroplasty…” As this procedure involved people, the ethical form number should be mentioned.

Table 1 should be re-written. Although used Mean (SD) CV, 2.90 (0.55), 19%, used (2.90 ± 0.55), 19 %.

All Latin terms, for example in vivo or in vitro, should be in italic font.

“We found an overall statistically significant difference between the surgeons in BM and that the CSA of the construct significantly impacted the BM. This finding collaborates findings of another study investigating the relationship between AdhFix plate thickness and biomechanical performance [7].” Then, what is the novelty of this research if others in the literature had already published about?

Conclusion should be separated from discussion.

More references should be included in the manuscript.

English should be improved in all manuscript.

Author Response

Response Letter

Dear reviewer

Thank you very much for your time and your comments. We, the authors, believe that the paper has now, after adding and editing according to your comments, improved significantly. In the following section, we reply to each of your comments (marked in red) and refer to specific edits in the paper. Thank you once again for taking the time and contributing to the enhancement of this scientific paper.

Comment 1: English should be improved in all manuscript.

Response 1: Thank you for your comment. We have carefully reviewed the paper again, making minor edits to address spelling, grammar, and sentence structure issues. We believe the revised version now meets the standards of appropriate academic English. Please note that two of the authors are native English speakers, and they have both edited the manuscript.

Comment 2: The title should be re-written to explore the main idea of this manuscript that is the statistical analysis based on reproducibility of handling of different surgeons in biomechanical analysis of AdhFix.

Response 2: Thank you for your comment. We have now altered the title of the paper to: ‘Biomechanical Variability and Usability of a Novel Customizable Fracture Fixation Technique’ (line 2-3). We would argue against using the term 'reproducibility' since the term 'variability' covers topics related to both 'reproducibility' and 'repeatability'.

Comment 3: The abstract should be re-written; it is very confusing. In addition, the novelty of this research is not so clear in the abstract.

Response 3: Thank you for your comment. We've revised the abstract to enhance clarity and emphasize the novelty of our research. Please refer to the abstract section in the draft (line 18-34).

Comment 4: Introduction: In general, the novelty of this research is not so clear. In addition, the authors should be emphasized the difference of this research from others performed with the same materials, as they afformed that “Up until now the biomechanical studies of the AdhFix technique have been analyzed 62 using only a single biomechanical researcher or a single surgeon to eliminate potential 63 variations and allow for a comparison between different osteosynthesis techniques and 64 AdhFix [7, 8]”

Response 4: Thank you for your comment. We have now placed more emphasis on the novelty of this study in the introduction-section. Additionally, we have provided elaborations on the distinctions between the studies previously published on this technique and the current draft. Please refer to the introduction-section in the draft (line 56-67, 78-79 and 83-90).

Comment 5: Materials and methods: “AdhFix is composed of trifunctional allyl and thiol triazine-trione monomers, a 78 photo-initiator (diphenyl (2,4,6-trimethylbenzoyl) phosphine oxide (TPO)) and hydroxy- 79 apatite. It is cured with a handheld high-energy visible (HEV) light lamp and anchored to 80 the bone with conventional metal screws [7].” The chemical structure of AdhFix should be included in the manuscript, this information is very interesting and explain some important aspects of mechanical behavior.

Response 5: Thank you for your comment. We have now included the chemical structure of the AdhFix composite (line 94-96).

Comment 6: “The fixture was mounted on a universal testing machine (Inspekt Duo 5, Hegewald & Peschke, Nossen, Germany) with a 5 kN load cell and a quasi-static compression rate of 3 mm/min (Figure 3).” Specific standard used in mechanical test should be mentioned.

Response 6: Thank you for your comment. A specific ASTM or ISO standard was not used in this test due to the novel nature of the fixation patch. However, we have provided a detailed description of the test setup so that others can replicate the tests. Please refer to methods-section (line 125-128).

Comment 7: “Six surgeons with various experience levels were recruited in this study: A consultant hand surgeon (surgeon 1), a specialist hand surgeon (surgeon 2), a specialist arthroplasty…” As this procedure involved people, the ethical form number should be mentioned.

Response 7: Thank you for your comment. We have now added ‘Institutional Review Board Statement’ and ‘Informed Consent Statement’ at the end of the draft to make the ethical considrations clear. Please refer to the final section of the draft (line 350-355).

Comment 8: Table 1 should be re-written. Although used Mean (SD) CV, 2.90 (0.55), 19%, used (2.90 ± 0.55), 19 %.

Response 8: Thank you for your comment. We agree with this remark and have now altered the table accordingly. Now the setup is as follows: mean ± SD (CV). Example: 2.01 ± 0.34 (18%). Please refer to table 1 (line 162-164).

Comment 9: All Latin terms, for example in vivo or in vitro, should be in italic font.

Response 9: Thank you for your comment. We agree and have now addressed the issue and consistently edited the Latin terms to be in italic font.

Comment 10: “We found an overall statistically significant difference between the surgeons in BM and that the CSA of the construct significantly impacted the BM. This finding collaborates findings of another study investigating the relationship between AdhFix plate thickness and biomechanical performance [7].” Then, what is the novelty of this research if others in the literature had already published about?

Response 10: Thank you for your comment. We have now edited this section since the point was not clear. Previously in the draft we addressed the main objectives and novelty in this study; the inter- and intra-surgeon biomechanical variability, which were investigated here for the first time. With this statement, as referred to in the comment, we aimed to convey that our additional discovery regarding the impact of cross-sectional area on bending moment aligns with the findings of a previous study. The novelty of our finding in this regard, lies in the fact that, in this study, we included the width (i.e., cross-sectional area) rather than just the thickness. Consequently, we provided further clarification on this topic in the draft. Please refer to the discussion-section (line 254-258).

Comment 11: Conclusion should be separated from discussion.

Response 11: Thank you for your comment. We agree and have now made a separate section for the conclusions.  

Comment 12

More references should be included in the manuscript

Response 12: Thank you for your comment. We've added additional references to support arguments in both the introduction and discussion sections.

Reviewer 2 Report

The article is interesting and well-written.

The fracture fixation technique is innovative and could lead to benefits in surgical treatments, potentially reducing post-operative risks and facilitating the functional recovery of patients, as highlighted in the introduction and discussion. Regarding the research design, the structuring of the article appears coherent and thorough. The use of tables and images explain in detail the materials, methods and results of the research.

The discussions are supported by the results and the explanation of the limitations of the study is precise.

In my opinion, an in-depth analysis of the "promising biochemical results" (line 54) is necessary. Explaining these results in detail regarding the state of the art would be able to make the advantages and properties of the technique under study more precisely understood.

A technical note: in line 97, "A consultant" should be written with the article "a" in the lowercase

Author Response

Response Letter

 Dear reviewer

Thank you very much for your time and your comments. We, the authors, believe that the paper has now, after adding and editing according to your comments, improved significantly. In the following section, we reply to your comments (marked in red) and refer to specific edits in the paper. Thank you once again for taking the time and contributing to the enhancement of this scientific paper.

Comment 1: The fracture fixation technique is innovative and could lead to benefits in surgical treatments, potentially reducing post-operative risks and facilitating the functional recovery of patients, as highlighted in the introduction and discussion. Regarding the research design, the structuring of the article appears coherent and thorough. The use of tables and images explain in detail the materials, methods and results of the research.

Comment 2: The discussions are supported by the results and the explanation of the limitations of the study is precise.

Comment 3: In my opinion, an in-depth analysis of the "promising biochemical results" (line 54) is necessary. Explaining these results in detail regarding the state of the art would be able to make the advantages and properties of the technique under study more precisely understood.

Response: Thank you for your comment. We agree, and have now elaborated on the previous biomechanical findings to make it more clear why this technique might be an adjuvant in fracture management. Please refer to the introduction section (line 56-67).

Comment 4: A technical note: in line 97, "A consultant" should be written with the article "a" in the lowercase

Response: Thank you for that comment. We agree, and have now corrected that error.

Round 2

Reviewer 1 Report

Ref. No.: bioengineering-2616626-v2

Subject: Decision on Manuscript: Variability and Usability of a Novel Customizable Fracture Fixation Technique: A Biomechanical Evaluation

Journal: Bioengineering

Dear Editor,

I would like to thank you for the invitation to collaborate to review process of article “Variability and Usability of a Novel Customizable Fracture Fixation Technique: A Biomechanical Evaluation”. My recommendation is described below:

The authors did the required all corrections and the manuscript is publishable in current version.

Ref. No.: bioengineering-2616626-v2

Subject: Decision on Manuscript: Variability and Usability of a Novel Customizable Fracture Fixation Technique: A Biomechanical Evaluation

Journal: Bioengineering

Dear Editor,

I would like to thank you for the invitation to collaborate to review process of article “Variability and Usability of a Novel Customizable Fracture Fixation Technique: A Biomechanical Evaluation”. My recommendation is described below:

The authors did the required all corrections and the manuscript is publishable in current version.